

# Decadal tropospheric ozone radiative forcing estimations with offline radiative modelling and IAGOS aircraft observations

Pasquale Sellitto[1,2], Audrey Gaudel[3,4], Bastien Sauvage[5].

[1]Univ. Paris Est Créteil and Université de Paris, CNRS, Laboratoire Interuniversitaire des Systèmes Atmosphériques, Institut Pierre Simon Laplace, Créteil, France
[2]Istituto Nazionale di Geofisica e Vulcanologia, Osservatorio Etneo, Catania, Italy
[3]CIRES, University of Colorado, Boulder, USA
[4]NOAA Chemical Sciences Laboratory, Boulder, USA
[5]LAERO, Laboratoire d'Aérologie, Université Toulouse III Paul Sabatier, CNRS, Toulouse, France

*Correspondence to*: Pasquale Sellitto (pasquale.sellitto@lisa.ipsl.fr)

**Abstract.** We use an offline radiative transfer model driven by IAGOS aircraft observations, to estimate the tropospheric ozone radiative forcing (RF) at decadal time scale (two time intervals between 1994-2004 and 2011-2016 or 2019), over 11 selected Northern Hemispheric regions. On average, we found a systematic positive trend in the tropospheric ozone column (TOC) for both time intervals, even if trends are reduced in 2019 ($\Delta$TOC +2.5±1.7 DU, +9.3±7.7%) with respect to 2011-2016 ($\Delta$TOC +3.6±2.0 DU, +14.9±11.5%). The reduced TOC average trend in 2019 with respect to 2011-2016, originates mostly from decreases of the lower tropospheric ozone column (LTOC) trends and limited variations for upper tropospheric ozone column (UTOC) trends, in the tropics. These average reductions in TOC trends are not accompanied with reductions of the tropospheric ozone RF, between 2011-2016 (4.2±2.4 mW m-2 per year) and 2019 (3.8±3.6 mW m-2 per year). This disconnection depends by the smaller RF sensitivity to LTOC than UTOC changes. Correspondingly, the total tropospheric ozone RF sensitivity varies between 18.4±7.4 mW m-2 per DU, in 2011-2016, and 31.6±20.3 mW m-2 per DU, in 2019. About 84-85% of the tropospheric ozone RF occurs in the longwave, with ~4-6% larger values of this proportion in the tropics than in the extra-tropics. Our estimates are 60-90% larger than the most recent global average tropospheric ozone RF estimates with online modelling. Our study underlines the importance of the evolution of ozone vertical profiles for the tropospheric ozone RF.

## 1 Introduction

Tropospheric ozone is a secondary atmospheric pollutant. It is either formed through photochemical reactions, which mostly occur in the boundary layer and involve primary anthropogenic pollutants, or is transported from the stratosphere (e.g. Seinfeld and Pandis, 1997). Besides being an air pollutant, with adverse effects on human health and the biosphere (e.g. Monks et al. 2015), tropospheric ozone is a strong short-lived greenhouse gas (Skeie et al., 2020). In addition to the dominant radiative effect in the longwave (LW) spectral range, tropospheric ozone absorbs also ultraviolet and visible radiation, thus it has a



radiative effect in the shortwave (SW) spectral range. Due to the progressive increase of anthropogenic emissions of tropospheric ozone precursors, such as nitrogen oxides and volatile organic compounds, the tropospheric ozone burden increased globally of up to 50% since early 1900s (e.g. Szopa et al., 2021). Despite ongoing regulations of the anthropogenic

emissions of its precursors, the tropospheric ozone burden was found to continuing increase well into the 2010s (Gaudel et al., 2020, hereafter referred to as G20). Tropospheric ozone increases since the preindustrial era are associated with a radiative forcing (RF). With an estimated present-day global average RF between $0.35 \, \text{W m}^{-2}$ (uncertainty range: $0.08–0.61 \, \text{W m}^{-2}$) (Skeie et al., 2020) and $0.47 \, \text{W m}^{-2}$ (uncertainty range: $0.24–0.70 \, \text{W m}^{-2}$) (Forster et al., 2021) since preindustrial era, tropospheric ozone is the third most important anthropogenic climate forcing agent, after carbon dioxide and methane.

As preindustrial era ozone distributions are not available in terms of observations, long-term tropospheric ozone RF estimations are obtained with modelling tools, based on hypotheses on preindustrial emissions and ozone burdens. A significant set of ground-based, ozone sondes and satellite observations are available for more recent times, which can be used to corroborate modelling results with RF estimations at decadal time scales. Thus, tropospheric ozone profiles can be derived with ozone sondes (e.g. Wang et al., 2024) and from ground-based instruments like Fourier transform infrared spectroscopy (e.g. Garcia

et al., 2022). Global-scale ozone profiles at the decadal time scale, including a tropospheric ozone, can be derived with satellites, which can subsequently be used as a source of information to estimate decadal tropospheric ozone RF. Ziemke et al. (2019) have shown, using satellite data over the period 1979-2016, that the tropospheric ozone column increased of up to 3 DU per decade, or more, even if with large regional inhomogeneities, which translates in a significant RF. More recently, Pope et al. (2024) have estimated the tropospheric ozone radiative effect with satellite data over the period 2008-2019 and shown

negligible trends resulting in a very limited decadal RF, in more recent periods. Another source of vertical-resolved ozone observations is the IAGOS (In-Service Aircraft for a Global Observing System) database (Petzold et al., 2015), which is based on ozone measurements by analysers operating on a large number of commercial aircraft flights worldwide. The observations-based RF estimations associated with decadal trends of tropospheric ozone need a radiative transfer tool to connect these trends to their radiative impacts. This can be provided by offline radiative transfer models or, more easily, pre-constructed radiative

forcing kernels (e.g. Maycock et al., 2021). While this latter approach can be easier to implement, full ad-hoc radiative transfer calculations assure more flexible RF estimations, and are able e.g. to catch the detailed impacts of background atmosphere and of specific vertical shape of the tropospheric ozone profiles.

In this paper, we first formalise a general approach for offline tropospheric ozone RF estimations associated with observed ozone trends, and then we apply this approach to tropospheric ozone trends of the IAGOS database at the 11 regions defined

by G20. In G20, we also presented RF for tropospheric ozone trends based on IAGOS observations, using a column conversion factor as a basic radiative forcing kernel. The present paper builds upon G20 but extends it in two main aspects: 1) full radiative calculations, considering ozone's vertical distribution, are used to obtain the decadal tropospheric ozone RF in the same period as G20 (1994-2004 versus 2011-2016), and 2) additional ozone trends and RF estimations are obtained by prolonging the analysis to the year 2019. Point 1 allows the specific attribution of the RF to specific vertical shape variabilities above the 11

regions (e.g. different trends in the upper and lower troposphere, UT and LT), and the differential study of the SW and LW





impacts. Point 2 allows the monitoring of the trends and RF impacts at the latest period before the COVID crisis, which had a specific impact on ozone precursors emission and ozone trends (e.g. Chang et al., 2022).

This paper is structured as follows. In Sect. 2 the data and methods used in this work are introduced, including the offline radiative transfer modelling framework. In Sect. 3 results are presented and discussed. In Sect. 4 conclusions are drawn.

## 2 Data and Methods

### 2.1 Offline radiative transfer modelling

The overall idea behind this work is to use an observational description of the decadal trends of tropospheric ozone as input of accurate radiative transfer calculations with a line-by-line offline radiative transfer model (RTM). Similar methodologies have been used in the past for aerosol studies (e.g. Sellitto et al., 2022, 2023) and deviate from the past radiative-kernel-based tropospheric ozone RF estimations. This approach allows the realistic description of the horizontal (regional) and vertical distribution of radiative forcing agents and their temporal evolution, through observations, and avoid the use of approximations of the radiative transfer problem through column parameterisations (like done by G20), or vertically-resolved radiative kernels (like done e.g. by Skeie et al., 2020, Pope et al., 2024).

The overall scheme of the offline RTM calculations used in this paper is shown in Fig. 1. Tropospheric ozone vertical profiles are taken from the IAGOS data base and averaged over three periods, 1994-2004, 2011-2016 and 2019, and the 11 regions defined by G20. Based on their latitude ranges, these average profiles are put in standard mid-latitude or tropical atmospheres taken from the AFGL (Air Force Geophysics Laboratory) data (Anderson et al., 1986). Clear sky conditions and a standard aerosol profile are used for all cases. These atmospheres are fed to the UVSPEC (UltraViolet SPECtrum) radiative transfer model in its libRadtran (library for Radiative transfer) implementation (Emde et al., 2016), operating at both the SW and LW spectral ranges. Spectra are simulated between 0.3 and 3.0 μm, for the SW range, and from 3.0 to 100.0 μm, for the LW range. We estimate the decadal tropospheric ozone instantaneous RF by comparing the radiative flux outputs at top of the atmosphere (TOA) using the 2011-2016 with the 1994-2004 tropospheric ozone average profiles, at all regions. This compares with the time frames of G20. Additional RF estimates are obtained by comparing 2019 and 1994-2004 averages.





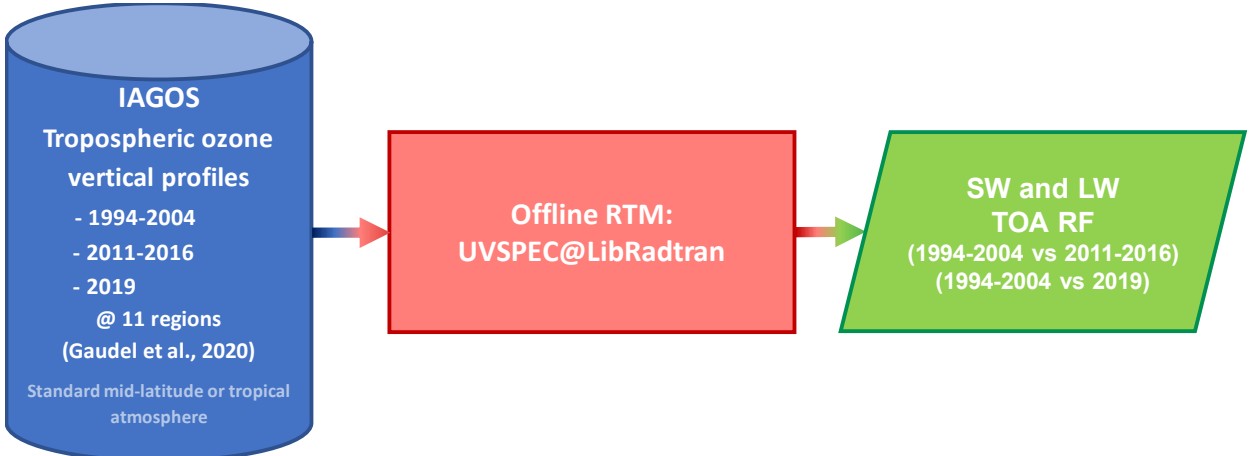

**Figure 1: Scheme of the offline radiative transfer modelling used in this work.**

## 2.2 IAGOS data

The European research infrastructure, In-service Aircraft for a Global Observing System (IAGOS, https://www.iagos.org/, last access November 2024), provides in situ measurements of chemical species on board several commercial fligths. Its predecessors, MOZAIC (Measurement of Ozone and Water Vapor by Airbus In-Service Aircraft: Marenco et al., 1998) and CARIBIC (Civil Aircraft for the Regular Investigation of the Atmosphere Based on an Instrument Container, e.g. Brenninkmeijer et al., 1999), relied on the same principle. The IAGOS infrastructure has been collecting high-quality continuous ozone concentration profiles up to about 12 km (~180 hPa) or less aboard commercial aircraft since 1994 (Blot et al., 2020). Ozone is measured using a UV analyser (Thermo Scientific, model 49) with a total uncertainty of ±2 nmol mol⁻¹ ±2% (Nédélec et al., 2015). For this study, we cluster the ozone profiles above 11 regions (listed here from north to south): Western Europe, Eastern North America, Western North America, Northeast China/Korea, Southeast US, Persian Gulf, Southeast Asia, South Asia (formerly known as India, e.g. in G20), Northern South America, Gulf of Guinea, Malaysia/Indonesia. The 50-hPa vertical resolution of the profiles is homogenised across the dataset. As done in G20, in this study, stratospheric fresh air masses characterized by ozone mixing ratio of 125 ppbv and above have been filtered out. Table 1 summarises the number of ozone profiles per region and for the three time-intervals used in the present study. It is important to notice that two of the regions defined above have only one profile (Southeast Asia) or none (Malaysia/Indonesia) available in 2019, and in general the 2019 time-period has significantly less profiles available than the other two time periods. Thus, comparisons at the specific regional scale with the time-period 2019 should be taken with caution. In the following, we will discuss this latter time-period only in terms of global and broad-regional (e.g. tropics and mid-latitudes) impacts. Based on the IAGOS vertical concentration profiles, partial ozone columns have been additionally calculated to investigate impacts of the tropospheric ozone variability at specific altitude ranges on the RF. In particular, lower tropospheric (LT), upper tropospheric (UT) and tropospheric (T) ozone columns have been obtained by integrating the concentration profiles from surface to 6 km, from 6 to 11 km and from surface to 11 km, respectively. We decided to keep the vertical intervals constant at all latitude



ranges, despite the varying troposphere depth and tropopause height, so to keep the analysis and the interpretation of results simple.


**Table 1: Number of IAGOS profiles per region for three time-intervals used in this study. The regions follow the same definition as in G20.**

|  | Abbreviation | N. of profiles 1994-2004 | N. of profiles 2011-2016 | N. of profiles 2019 |
|---|---|---|---|---|
| Western Europe | W Eu | 19281 | 7047 | 770 |
| Eastern North America | E NAm | 6536 | 1734 | 78 |
| Western North America | W NAm | 613 | 384 | 52 |
| Northeast China/Korea | NE Ch/Ko | 2746 | 1207 | 71 |
| Southeast US | SE US | 2950 | 359 | 87 |
| Persian Gulf | PeGul | 1184 | 1179 | 230 |
| Southeast Asia | SE As | 929 | 1458 | 15 |
| South Asia | S As | 468 | 416 | 1 |
| Northern South America | N SAm | 960 | 465 | 17 |
| Gulf of Guinea | GulGu | 988 | 751 | 239 |
| Malaysia/Indonesia | Ma/In | 159 | 401 | 0 |
| Total |  | 36814 | 15401 | 1560 |

## 3 Results

### 3.1 Tropospheric ozone decadal changes

Figure 2 shows average tropospheric ozone concentration profiles in the different regions addressed in this study, in the time-periods 1994-2004, 2011-2016 and 2019. The average LT, UT and T ozone columns (LTOC, UTOC and TOC), associated with these regions and time-intervals (corresponding with Fig. 2 profiles) are shown in Fig. 3, and the percent differences of the average 2011-2016 (panel a) and 2019 (panel b) with respect to the average columns 1994-2004 are shown in Fig. 4. Global

average values of LTOC, UTOC and TOC variations (ΔLTOC, ΔUTOC and ΔTOC, as absolute and percent values) are reported in Tab. 2. To be consistent with the 2019 ending period, we also calculated the global average values ΔLTOC, ΔUTOC and ΔTOC, for 2011-2016 versus 1994-2004, by excluding Malaysia/Indonesia, because there aren't any ozone observations in Malaysia/Indonesia in 2019 (Tab. 1).







**Figure 2: Ozone concentration profiles in the 11 regions listed in Tab. 1, averaged over the periods 1994-2004 (black lines), 2011-2016 (orange lines) and 2019 (red lines).**

Consistently with G20, as a general decadal trend, TOC increased worldwide by +3.6±2.0 DU (+14.9±11.5%), in 2011-2016 with respect to IAGOS averages in 1994-2004 (see Tab. 2). The increasing trend is more pronounced for LTOC (+2.8±1.8 DU, +17.7±16.0%), than UTOC (+0.9±0.7 DU, +9.9±9.3%). The increasing decadal trend for TOC is confirmed for the estimations in 2019 with respect to 1994-2004. On average, while UTOC increases are confirmed as well, in 2019 (+0.9±1.5 DU, +9.3±18.8%), LTOC increases are ~50% smaller in 2019 (+1.6±1.1 DU, +9.4±7.5%) in comparison with the trends between 2011-206 and 1994-2004. As a consequence, the overall increase of TOC, with respect to the time interval 1994-2004, is smaller in 2019 (+2.5±1.7 DU, +9.3±7.7%) than in 2011-2016. This seems to point at a general LT-driven decrease of the decadal ozone increasing trends, comparing the year 2019 with the period 2011-2016, which is likely explained by a





continuous reduction in the emissions of the primary anthropogenic ozone-precursor pollutants in the Northern Hemisphere. The effect of the implementation of emission reduction policies for ozone precursors, such as nitrogen dioxide, carbon monoxide and volatile organic compounds (VOCs), on the surface and tropospheric columns ozone was recently discussed (Elshorbany et al., 2024).

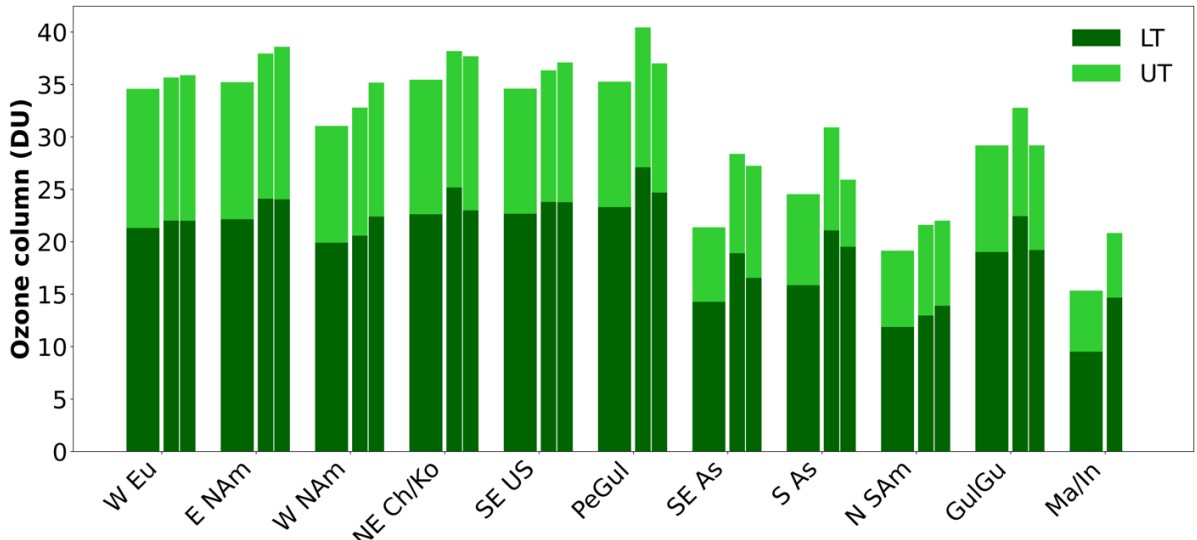

**Figure 3: Average LT (dark green bars) and UT ozone columns (stacked light green bars) in the periods 1994-2004 (large bars on the left), 2011-2016 (bars at the centre) and 2019 (bars on the right), for each of the 11 regions listed in Tab. 1.**

Looking at specific regions, the largest slowdown of the TOC decadal increases seen when we compare the end time-periods 2019 and 2011-2016 is observed in the tropical regions of South Asia, Persian Gulf and the Gulf of Guinea and, to a smaller extent, Southeast Asia. The more pronounced reduction in the LTOC trends, with respect to the UTOC trends, is clearly visible

over North East China/Korea (LTOC trends vary from +11.3%, in 2011-2016, to +1.6%, in 2019; UT trends vary from +1.4%, in 2011-2016, to +14.5%, in 2019) and Southeast Asia (LTOC trends vary from +32.6%, in 2011-2016, to +16.1%, in 2019; UT trends vary from +33.0%, in 2011-2016, to +50.2%, in 2019). In these example regions, the LTOC levels, in 2019, are almost back to the 1994-2004 levels, while the UTOC continues its increase between 2011-2016 and 2019. These UTOC increases are visible starting from about 7-8 km altitudes (see Figs. 2d and g). Trajectories of evolution of LT and UT are

different over mid-latitude regions. Above Western Europe and North America, we observe a larger positive TOC trend in 2019 than in 2011-2016, driven by LTOC trends. As an example, in Western North America, the LTOC increased by +3.5%, in 2011-2016, and of +12.5%, in 2019, with respect to 1994-2004, with LTOC increases distributed at all LT altitudes (Fig. 2c). More vertically-localised LTOC increases at lower altitudes are visible for Eastern North America, Southeast US and Western Europe regions (Figs. 2b, e and a, respectively). A general increase of the positive decadal ozone trend, in 2019 with

respect to 2011-2016, is observed at lower altitudes, over mid-latitude regions except for the Northeast China/Korea region.




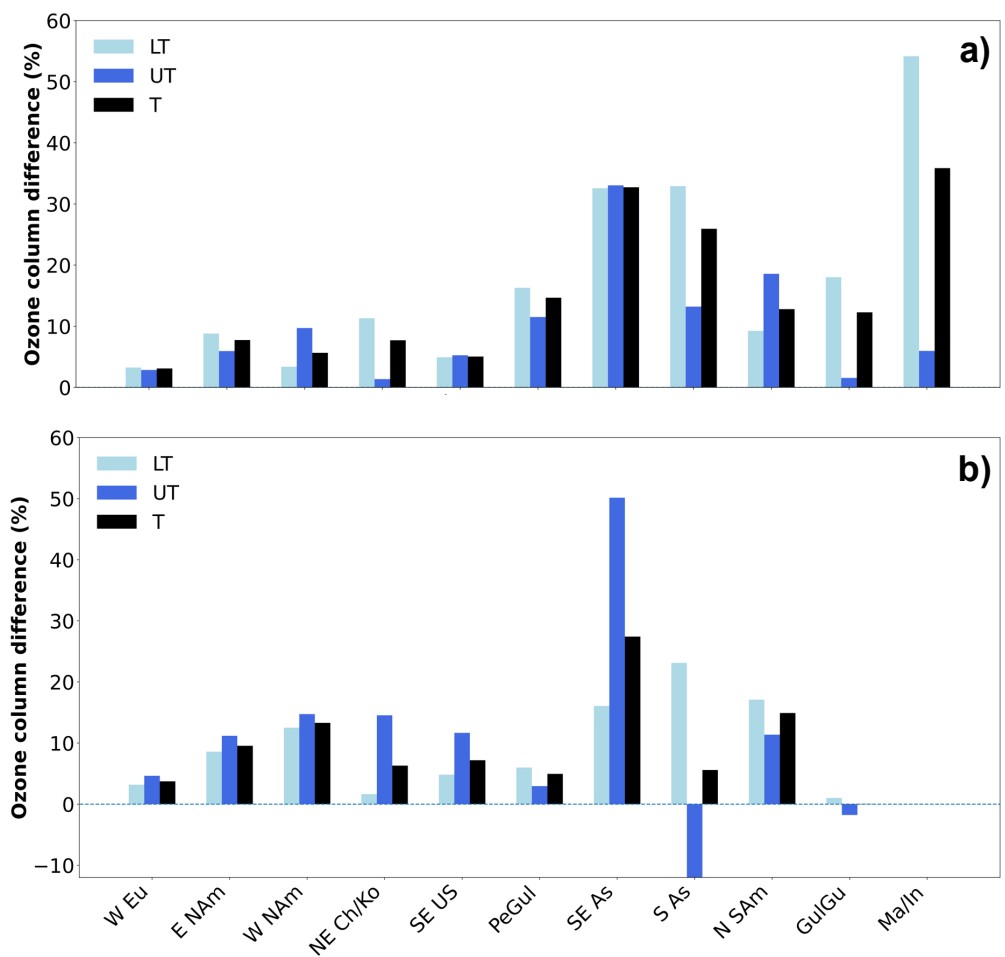

**Figure 4: LT (sky blue bars), UT (blue bars) and total tropospheric ozone percent difference (black bars), for each of the 11 regions listed in Fig. x, for the periods 2011-2016 (panel a) and 2019 (panel b) with respect to the period 1994-2004.**

**Table 2: Average total ozone column (ΔTOC), lower tropospheric column (ΔLTOC) and upper tropospheric column (ΔUTOC) difference (in DU and percent), for the periods 2011-2016 (upper column) and 2019 (lower column) with respect to the period 1994-2004. The values in parenthesis are obtained excluding Malaysia/Indonesia.**

|  | ΔTOC (DU) | ΔTOC (%) | ΔLTOC (DU) | ΔLTOC (%) | ΔUTOC (DU) | ΔUTOC (%) |
|---|---|---|---|---|---|---|
| 2011-2016 | 3.6±2.0 (3.5±2.0) | 14.8±11.5 (12.8±11.5) | 2.8±1.8 (2.5±1.7) | 17.7±16.0 (14.1±11.0) | 0.9±0.7 (0.9±0.7) | 9.9±9.3 (10.3±9.7) |
| 2019 | 2.5±1.7 | 9.3±7.7 | 1.6±1.1 | 9.4±7.5 | 0.9±1.5 | 9.3±18.8 |



### 3.2 Radiative forcing

Figure 5 shows our offline radiative calculation of the decadal RF associated with the trends in TOC discussed in Sect. 3.1, for the 11 regions. Averages over the 11 regions are summarised in Tab. 3. The SW and LW components of the tropospheric ozone RF are shown separately, and both the RF for the trends calculated using 2011-2016 and 2019 ending periods, with respect to 1994-2004, are shown in the figures. The average total (SW+LW) decadal tropospheric ozone RF is 60.4±34.8 mW m$^{-2}$, for the 2011-2016 period, and 75.7±72.2 mW m$^{-2}$, for the 2019 period. Considering the different temporal interval for the

two trends estimations (14.5 years, for 2011-2016 average, and 20 years, for 2019 average), this corresponds to very similar RF per year, i.e. 4.2±2.4 mW m$^{-2}$ per year, for 2011-2016, and 3.8±3.6 mW m$^{-2}$ per year, for 2019. Thus, despite the general decreasing positive trends, when considering 2019 with respect to 2011-2016, the RF per year is not significantly decreasing, on average, in 2019 with respect to 2011-2016. This can be readily linked to the vertical region where reductions in trends are coming from. As discussed in Sect. 3.1, in general, the reduction in positive trends in 2019 with respect to 2011-2016 is mainly

linked to a decrease in the LTOC positive trends, while the UTOC trends stayed approximately constant. The instantaneous TOA RF due to tropospheric ozone is much more sensitive to the ozone changes in the UT than in the LT (e.g. Worden et al. 2011). As an example, in Southeast Asia, the TOC positive trend decreased from 2011-2016 to 2019, with respect to 1994-2004, mostly linked to a reduction of LTOC trends and little changes in UTOC. The associated tropospheric ozone RF per year stayed approximately constant between 2011-2016 and 2019. Our results suggest that local emission reduction for ozone

precursors, leading to decreases in ozone levels at lower altitudes, while beneficial for air quality, health and the biosphere, might not lead to a clear decrease in short-term radiative impacts, hence climate. Another important result of our tropospheric ozone RF estimates is that the uncertainty of our decadal trends is increasing, possibly due to the observed divergence in LT to UT trajectories in tropical and mid-latitudes regions, as discussed in Sect. 3.1, or for the significantly smaller number of observations in 2019 with respect to 2011-2016.

Our average estimates cannot be directly compared with online model global average tropospheric ozone RF estimates but can still be used as a reference where IAGOS observations are available (thus mostly the Northern Hemisphere). The most recent global tropospheric ozone RF estimate available is the one provided by Skeie et al. (2020). They provided an estimate of 0.35 W m$^{-2}$ (with an uncertainty range of 0.08-0.91 W m$^{-2}$), for a time-interval of 160 years since the pre-industrial era, which translates to 31.7 (7.3-82.5) W m$^{-2}$, for an equal period of 14.5 years as out 2011-2016 estimates, and 2.2 (0.5-5.7) mW m$^{-2}$

per year. This is very similar to the previous estimate of 2.4±1.2 mW m$^{-2}$ per year, obtained by Myhre et al. (2017) and used as reference in G20. Our average values are ~60 to 90% larger than previous global average estimates with online models.





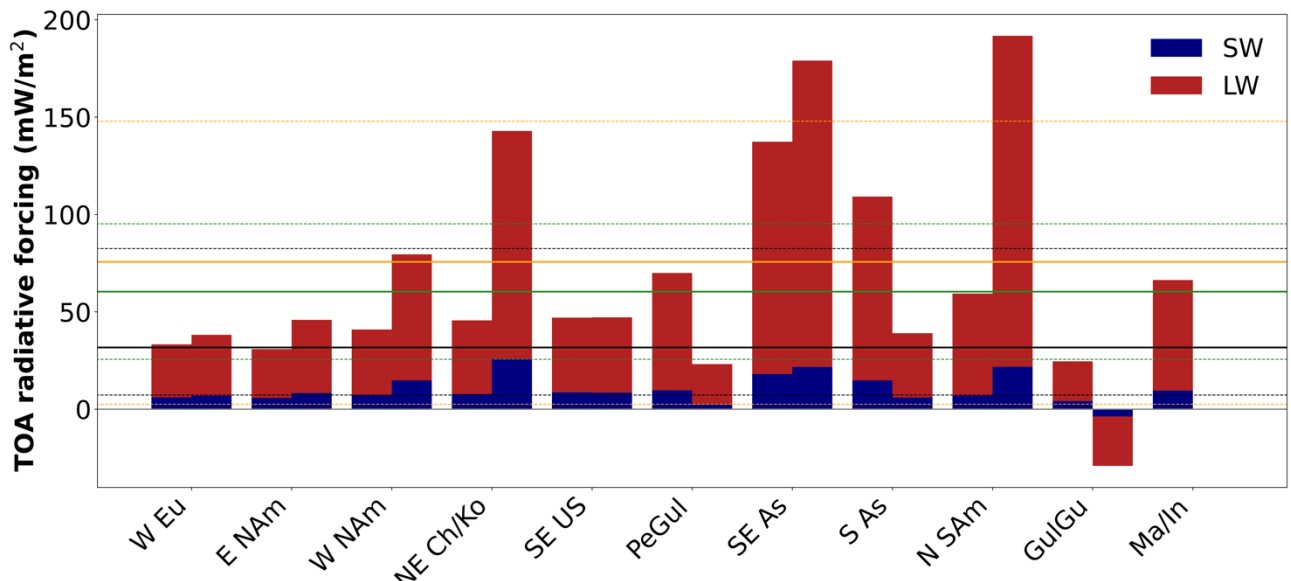

**Figure 5: SW (blue bars) and LW (stacked red bars) decadal tropospheric ozone instantaneous RF, for each of the 11 regions listed in Tab. 1, for the periods 2011-2016 (bars on the left) and 2019 (bars on the right) with respect to the period 1994-2004.**
**Horizontal lines represent the overall 2011-2016 (solid green line) and 2019 (solid yellow line) averages, and the global average RF of Skeie et al. (2020) scaled to 14.5 years (solid black line). Horizontal dotted lines represent the uncertainty range of respective solid lines.**

Our detailed radiative calculations allow the estimation of the separate SW and LW tropospheric ozone RF and their ratio, associated with different vertical distributions of ozone concentration and their decadal trends. Table 3 summarises the SW

and LW RF for our estimations and Tab. 4 the ratio of the LW to total (SW+LW) RF ratio. The LW contribution to total tropospheric ozone RF is ~84±3% (max: 88%, min: 81%), in 2011-2016, and ~85±3% (max: 89%, min: 81%), in 2019. As also observed by Doniki et al. (2015) and Gaudel et al. (2024), the LW radiative effect of tropospheric ozone is larger in the tropics than extra-tropics, by 4.2±1.8%, in 2011-2016, and by 5.6±1.8%, in 2019.

**Table 3: SW, LW and total tropospheric ozone RF, for the periods 2011-2016 (upper column) and 2019 (lower column) with respect to the period 1994-2004. The values in parenthesis are obtained excluding Malaysia/Indonesia.**

|  | SW RF (mW m$^{-2}$) | LW RF (mW m$^{-2}$) | Total RF (mW m$^{-2}$) | Total RF per year (mW m$^{-2}$ per year) |
|---|---|---|---|---|
| 2011-2016 | 9.0±4.1 <br> (8.9±4.3) | 51.4±30.8 <br> (50.9±32.4) | 60.4±34.8 <br> (59.8±36.6) | 4.2±2.4 <br> (4.1±2.5) |
| 2019 | 11.2±9.5 | 64.5±63.2 | 75.7±72.2 | 3.8±3.6 |




**Table 4: LW to SW tropospheric ozone RF ratio for 2011-2016 and 2019, with respect to 1994-2004, calculated for all regions, and extra-tropical and tropical regions. The values in parenthesis are obtained excluding Malaysia/Indonesia.**

|  | LW/(SW+LW) RF ratio All (%) | LW/(SW+LW) RF ratio Extra-tropical (%) | LW/(SW+LW) RF ratio Tropical (%) | Difference Tropical/Extra-tropical (%) |
|---|---|---|---|---|
| 2011-2016 | 84.2±2.5 (84.1±2.6) | 82.0±0.7 | 86.1±1.6 (86.2±2.8) | 4.2±1.8 (4.2±2.2) |
| 2019 | 84.7±3.2 | 81.9±0.3 | 87.5±1.8 | 5.6±1.8 |

We also calculated the RF sensitivity to TOC trends, in both the SW and LW, as well as for total SW+LW RF, as described in

Eq. 1-3. The average RF sensitivities, over the 11 stations, for both the 2011-2016 and 2019 ending periods, are summarised in Tab. 5. Our results show a significant change in sensitivity between the two ending periods, with average total (SW+LW) RF sensitivities of 18.4±7.4 W m$^{-2}$ per DU, in 2011-2016, and 31.6±20.3 W m$^{-2}$ per DU, in 2019. The larger RF sensitivity $\chi_{tot}$ for 2019 than 2011-2016 is likely associated with the observed changes in vertical distribution and their decadal trends, i.e. progressively relatively more UT and less LT ozone levels in 2019 than 2011-2016.

$$\langle \chi_{SW} \rangle = \langle \frac{SW\ RF}{\Delta TOC} \rangle \,, \tag{1}$$

$$\langle \chi_{LW} \rangle = \langle \frac{LW\ RF}{\Delta TOC} \rangle \,, \tag{2}$$

$$\langle \chi_{tot} \rangle = \langle \frac{SW\ RF + LW\ RF}{\Delta TOC} \rangle \,, \tag{3}$$

**Table 5: Average RF sensitivity to tropospheric ozone column change for SW ($\chi_{SW}$), LW ($\chi_{LW}$) and both ($\chi_{tot}$). The values in parenthesis are obtained excluding Malaysia/Indonesia.**

|  | $\langle \chi_{tot} \rangle$ (mW m$^{-2}$ DU$^{-1}$) | $\langle \chi_{SW} \rangle$ (mW m$^{-2}$ DU$^{-1}$) | $\langle \chi_{LW} \rangle$ (mW m$^{-2}$ DU$^{-1}$) |
|---|---|---|---|
| 2011-2016 | 18.4±7.4 (19.1±7.5) | 2.9±1.4 (3.1±1.5) | 15.5±6.1 (16.0±6.2) |
| 2019 | 31.6±20.3 | 4.8±3.1 | 26.8±17.6 |


## 4 Conclusions

In this paper, we have used the LibRadtran/UVSPEC offline radiative transfer model to estimate decadal tropospheric ozone instantaneous RF of tropospheric ozone trends, based on an observational description from IAGOS aircraft measurements.




Based on previous work of G20, these decadal RF are estimated over 11 regions, covering northern-hemispheric mid-latitudes
and tropics. Two time-interval are considered for the decadal trends in this work: 2011-2016 versus 1994-2004 averages (as
in G20), and 2019 versus 1994-2004 averages. As in G20, we have found a systematic global average positive trend, in the
Tropospheric Ozone Column (TOC), Lower Tropospheric Ozone Column (LTOC) and Upper Tropospheric Ozone Column
(UTOC), for the decadal ozone trends estimated in 2011-2016 versus 1994-2004. We found that the decadal TOC trends
calculated in 2019 is lower with respect to the 2011-2016 trends, with a ΔTOC of +2.5±1.7 DU (+9.3±7.7%), in 2019, and
+3.6±2.0 DU (+14.9±11.5%), in 2011-2016. The trends decrease seems to be driven by a general decrease of TOC in the
tropics. The tropical TOC decrease is driven by a decrease of LTOC trends, accompanied by a small increase in the UTOC
trends. On the contrary, the mid-latitude TOC trends are larger in 2019 than 2011-2016. At mid-latitudes, most of this increase
of TOC trend seems to be associated with an increase of the LTOC trend from 2011-2016 to 2019. We found that the decrease
in TOC trends are not accompanied with a reduction of the radiative forcing (RF) due to tropospheric ozone between 2011-
2016 (60.4±34.8 mW m$^{-2}$, 4.2±2.4 mW m$^{-2}$ per year) and 2019 (75.7±72.2 mW m$^{-2}$, 3.8±3.6 mW m$^{-2}$ per year). This is linked
to the much smaller sensitivity of the tropospheric ozone RF to the LTOC. We estimated such sensitivity through the $\chi_{tot}$
parameter, which is defined as the decadal tropospheric ozone RF per unit variation in TOC in a given time interval. The $\chi_{tot}$
parameter varied between 18.4±7.4 W m$^{-2}$ per DU, in 2011-2016, and 31.6±20.3 W m$^{-2}$ per DU, in 2019, likely associated
with the UTOC increase and the LTOC decrease between 2019 and 2011-2016. Our average estimates over the 11 regions are
~60-90% larger than previous global average estimates with online models, such as the latest estimate of Skeie et al. (2020)
(2.2 mW m$^{-2}$ per year, on average). The longwave (LW) contribution to total shortwave and longwave (SW+LW) RF is ~84-
85%, with ~4-6% larger values in the tropics than in the extra-tropics.

**Author contributions**

PS designed the study and realised the offline RF estimations. AG and BS provided the input IAGOS data. All authors
participated to the discussion of the results. PS wrote the manuscript and all authors contributed to its revision and editing.

**Competing interests**

The authors don't have competing interests to declare.

**Acknowledgements**

The authors acknowledge the support of the European Commission, Airbus and the airlines (Lufthansa, Air France, Austrian
Airlines, Air Namibia, Cathay Pacific, Iberia and China Airlines so far) who have carried the IAGOSCore equipment and



performed the maintenance since 1994. In its last 10 years of operation, IAGOS-Core has been funded by INSU– CNRS (France), Météo-France, Université Paul Sabatier (Toulouse, France) and Forschungszentrum Jülich (FZJ, Jülich, Germany). IAGOS has been additionally funded by the EC projects IAGOS-DS and IAGOS-ERI. The IAGOS-Core and the IAGOS-CARIBIC database are supported by AERIS. Audrey Gaudel's contribution was supported by the NOAA cooperative agreement with CIRES (NA17OAR4320101 and NA22OAR4320151).

## Financial support

This research has been supported by the Centre National d'Etudes Spatiales (grant no. TOTICE).

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
