# Peer review of "Decadal tropospheric ozone radiative forcing estimations with offline radiative modelling and IAGOS aircraft observations"

_EGUsphere, 2024_

## Author Comment (AC1)

**Reply to reviewers of the manuscript "Decadal tropospheric ozone radiative forcing estimations with offline radiative modelling and IAGOS aircraft observations", Sellitto et al.**

Dear Editor, dear anonymous Reviewers #1-2,

Many thanks for your constructive criticism and the very useful comments. Based on your comments, we have thoroughly revised our manuscript.

As a general point, and as a consequence of the convergence of both reviewers on this aspect, we have decided to extend the temporal analysis in two ways:

1) We have extended the second ending point to 2017-2019 (previously 2019 only). This is intended to increase the number of observations for this period, which was less than desirable when using the year 2019 only (see Reviewer #1's GC3 and Reviewer #2's GC4).

2) We have added a COVID and post-COVID more recent time interval, using a new ending period 2020-2023 (see Reviewer #1's GC8). This allows the analysis of the effect of the COVID crisis in the tropospheric ozone trends and RF, which is also studied, with different methods, in a number of publications recently.

The modification of the 2017-2019 (formerly 2019) time interval significantly changed the interpretation of the results and confirmed that the poor sampling obtained with the 2019 year only was misleading. The addition of the new 2020-2023 period confirmed the directions shown by the study of the period 2017-2019. We now see a clear stagnation of the TOC variation with respect to the same 1994-2004 starting period and an increasing reduction of the upper tropospheric ozone levels, which is not dissimilar to what show in other studies (now cited in our manuscript). Correspondingly, the decadal RF due to tropospheric ozone trends, is clearly decreasing from 2011-2016 to 2017-2019 to 2020-2023, and the COVID and post-COVID effects are visible in this latter period. This is now discussed in the revised version of the manuscript. We warmly thank the two Reviewers for suggesting us these new analyses, which, we think, made our paper much better than the previous version. Please note that most figures are changed to include the new time interval.

Please find more details and a point-by-point reply to the Reviewers' specific comments in the following (Reviewers' comments are in black and our replies in blue). We think that, thanks to your comments and suggestions, the present version of the manuscript is greatly improved with respect to the previous version.

Thank you very much,
Pasquale Sellitto on behalf of all co-authors

Radiative forcing estimations based on IAGOS mean tropospheric ozone profiles for different regions are provided with offline radiative modelling, enabling to assess the impact of the variability of the vertical distribution of tropospheric ozone on the radiative forcing. The radiative forcing calculations are important and follow a novel approach, that permits to make the distinction between shortwave and longwave RF for the different regions.

**General comments**

The manuscript deserves publication in ACP, but there is a need for further specifications or more details on the dataset used, some results (obtained values) should be better explained in comparison with other, previous studies, and some sensitivity analyses could be additionally performed.
We would like to thank Reviewer #1 with this series of questions and comments that help us clarified some important points of the paper. Please find our response below.

GC1) The manuscript builds further on the Gaudel et al., 2020 (named G20) study, but some more details of interest for the analysis described here should be given: what were the selection criteria for defining the 11 regions (see Table 1)?
We selected the 11 regions based on the IAGOS datasets providing a consistent number of data/observations to compute trends from 1994-2016, as done in Gaudel et al. (2020). We kept the same regions to expand the study for consistency.

GC2) Are the observations spatially representative for the defined region? Also, in contrast to the G20 study, in which tropospheric ozone trends are calculated based on Quantile Regression on monthly anomalies, tropospheric ozone decadal changes in this manuscript are estimated from the mean tropospheric ozone profiles for different periods, as shown in Figure 2. However, those mean tropospheric ozone profiles might be very dependent on the spatial and temporal distribution of the IAGOS observations over the region or over the time period. For instance, one period might be dominated by summertime observations, while the other period is mainly characterized by wintertime flights. Or, the large majority of the flights might be situated in the beginning of a time period for one region, but at the end of the time period for another region, making the comparison between the regions less meaningful. Also, during the early time period, most profiles might be originating from take-off/landing at the west side of the region, for instance, but on the east side of the region for one of the later periods.
In the supplemental material we are now adding the maps of the flight tracks and histograms of the monthly distribution of observations for each region and each time period (Figures S1 and S2 below). We note that the sampling in each region can vary in the overall number of flights depending on the time period (see revised Table 1 below) but the spatial (Figure S1) and temporal (Figure S2) coverage remain approximately the same except for some regions in the time period 2020-2023. To make this clear, we added the following sentences in the text (section 2.2): "The maps of the flight tracks and the histograms of the number of observations per month, for each region and each time period, are shown in Figs. S1 and S2, respectively. The density of the sampling in each region vary in terms of the number of flights depending on the time period (see Table 1) but the spatial (Fig. S1) and annual (Fig. S2) coverage is similar for the 11 regions, except for some regions for the time period 2020-2023. Notably, some discrepancies in spatiotemporal sampling, in comparison with the other time periods, are found for 2020-2023, which may introduce additional uncertainties in the final radiative forcing calculation. Most changes are for Western North America, Northern South America, Southeast Asia and Malaysia/Indonesia

regions. In addition, Malaysia/Indonesia has significantly less profiles available than the other regions. Thus, the analyses for these regions in 2023 should be taken with caution."

[Figure]

**Figure S1.** Flight tracks in the eleven regions of the study for the four time periods 1994-2004, 2011-2016, 2017-2019 and 2020-2023.

[Figure]

**Figure S2.** Histogram of the months for the eleven regions (eleven panels) and for the four time periods 1994-2004 (blue), 2011-2016 (orange), 2017-2019 (green) and 2020-2023 (purple).

**Table 1:** Number of IAGOS profiles per region for three time-intervals used in this study. The regions follow the same definition as in G20.

|  | Abbreviation | N. of profiles 1994-2004 | N. of profiles 2011-2016 | N. of profiles 2017-2019 | N. of profiles 2020-2023 |
|---|---|---|---|---|---|
| Western Europe | W Eu | 19101 | 7043 | 2998 | 2739 |
| Eastern North America | E NAm | 6536 | 1732 | 356 | 600 |
| Western North America | W NAm | 587 | 379 | 584 | 175 |
| Northeast China/Korea | NE Ch/Ko | 2747 | 1182 | 603 | 366 |
| Southeast US | SE US | 2934 | 359 | 201 | 184 |
| Persian Gulf | PeGul | 1184 | 1174 | 761 | 524 |
| Southeast Asia | SE As | 932 | 1442 | 757 | 252 |
| South Asia | S As | 469 | 427 | 198 | 150 |
| Northern South America | N SAm | 961 | 465 | 230 | 110 |
| Gulf of Guinea | GulGu | 993 | 750 | 829 | 540 |
| Malaysia/Indonesia | Ma/In | 159 | 400 | 77 | 73 |
| Total |  | 36603 | 15353 | 7594 | 5713 |

GC3) On top of that, there is clear temporal sampling difference between the two earlier periods and the year 2019, which will impact the mean tropospheric ozone profiles over the region as well. The impact of possible differences of the spatial and temporal sampling on the different mean tropospheric ozone profiles should, as a consequence, at least be mentioned or even better, somewhat assessed.

We decided to extend the third period to 2017-2019. This produced a better balance in the populations of the three periods (see Tab. 1). Please see the general introduction of this "reply to reviewers" document for more details on the broad implications of this choice.

GC4) Related to this, I would expect to see also the standard deviations of the mean tropospheric ozone profiles included in Fig. 2, in the average LTOC and UTOC in Figure 3, and in the LT, UT, and T ozone percent differences in Figure 4. Only the uncertainties for the worldwide TOC, LTOC and UTOC differences are provided in the text (page 6) and in Table 2, but it is not mentioned how these uncertainties are obtained (statistical mean over the different regions I assume?).
We tried different solutions and layouts to show the uncertainties for each region in Figs. 2-4 but this systematically raised severe readability issues with these figures. We still think that the global uncertainty of Table 2 (as the Reviewer says: it is the statistical mean over the regions, thus describing the variability of the trends) is the most important information, from this point of view.

GC5) Based on the standard deviations of the mean tropospheric ozone profiles in Fig. 2, one could perform a sensitivity analysis of the RF estimations on the input mean tropospheric ozone profile for each region. Given the comment on how spatial and temporal representative the mean tropospheric ozone profiles for each region are, this RF estimation sensitivity analysis would add an extra feature to your findings.
This is a good suggestion, but we feel that adding a specific RF estimation sensitivity study would: 1) be redundant with respect to the similar discussion later in our manuscript about the RF sensitivity $\chi_{tot}$, 2) be redundant with most works on radiative kernels, like Skeie et al. 2020 (cited in our manuscript), 2) render our manuscript less readable.

GC6) The obtained (global) RF estimates are compared with previous studies, but not with the values obtained in G20 (Fig. 6) for exactly the same regions, and one of your 2 periods, but with a different method. Why is this comparison not been made? I found this rather strange.
The estimations in G20, averaged over the same 11 regions, give a RF of about 80 mW/m$^2$, which is comparable, even if larger, to our ~60 mW/m$^2$. We briefly mention this in the revised text.

GC7) It also turns out that your average values are 60 to 90% larger than previous global average estimates with online models, but no explanations for this rather large offset have been given. The authors should go more in depth on this.
Based on our new RF estimations with the 2017-2019 extended period (2019 in the previous iteration) and the new 2020-2023 period, we have found that our overestimations of previous global averages are greatly decreased (now ~25% for 2017-2019 and even an underestimation of ~30% for 2020-2023). While this is clearly do to the marked decrease in TOC decadal trends more recently, another possible reason for deviations between our results and the ones from online models is the fact that we are limited to mostly the Northern Hemisphere (this is mentioned in the manuscript).

GC8) As many studies in the TOAR Special Issue pointed out, there was a decrease of tropospheric ozone column amounts during the COVID-19 period (and still continuing today). Have the authors not considered to quantify the impact of this effect on the RF forcing estimations by including a more recent year(s) than 2019 in their analysis? The authors should make reference to this (post-)COVID impact on tropospheric ozone and comment on their choice.
To gather more insights into this, we decided to extend our study to a fourth (post-COVID) period, 2020-2023. ). Please see the general introduction of this "reply to reviewers" document for more details on the broad implications of this choice.

**Specific comments**

SC1) Line 13: add "the year 2019"
This is now no more of interest, due to the modification of the time intervals of this study

SC2) Line 45: remove "a" before "tropospheric ozone"
SC3) Line 109: have additionally been
Both done

SC4) Line 137: "2016" instead of "206"
Done

SC5) Lines 156-157: Just to give an example of my previous comment on the spatial or temporal sampling: How confident are you that the higher LTOC increase for Western North America in 2019 (+12.5%) than in 2011-2016 (+3.5%) compared to 1994-2004, is not due to the fact that the 2019 sample is dominantly made up by summertime months, compared to the 2011-2016 sample?
This is now no more of interest, due to the modification of the time intervals of this study

SC6) Lines 186-189: Where do I have to note that the uncertainty of the decadal trends is increasing? In Table 3? But these are uncertainties over the regions, right? And also the trends themselves are increasing. Please clarify these statements.
We corrected "uncertainty" with "regional variability".

SC7) Line 194: 31.7 mW m-2 instead of W m-2
Done

SC8) Line 196: Try to give an explanation for this finding.
Please see our reply to GC7.

**Reviewer #2**

The study by Sellitto et al., titled "Decadal tropospheric ozone radiative forcing estimations with offline radiative modelling and IAGOS aircraft observations" uses aircraft observations and an off-line radiative transfer model (RTM) to investigate decadal ozone radiative forcing (RF). Overall, this is an interesting study within the scope of ACP. However, more details on the methods are required before it can be accepted for publication in ACP.

We would like to thank Reviewer #2 for the series of comments that help us improve the message of the paper. Please find our answers below.

**General Comments:**

GC1) The IAGOS forms a core part of the analysis in this paper, however, there is no discussion on the quality of the data used. How was the data filtered for anomalous or spurious values/profiles. Secondly, the data is averaged into regions (e.g. Figure 2 vertical profiles) but there is no information on the variability. I think it would be useful to add some information on this. Also, some information on how the regions are defined would be useful (e.g. map with regions shown etc.)

In the method section, we cite previous and thorough studies regarding the quality of the data. In the supplemental material we are now adding the maps of the flight tracks and histograms of the months for each region and each time period (Figures S1 and S2 below). Please see discussion for Reviewer #1's GC2.

GC2) The presentation of Figures 3 and 5 needs to be improved. It is not easy to see the difference between the vertical bars for the different time periods. I would suggest keeping the same colours but maybe changing the colour fill (e.g. hatching or dots etc.) for some of the bars.

We modified Fig. 3, which now identifies with a specific colour-contour each time period (also considering that we added a fourth one, 2020-2023. These colours are now consistent with Fig. 5, i.e. each average value carries the colour of the period identified in Fig. 5. We think that these modifications greatly improved the readability of our figures (which carry a lot of information each, we agree on that).

GC3) When you are calculating the RF, you are using the aircraft profiles and the RTM to calculate the radiative effect (RE) and then taking the different between two time periods (my interpretation anyway). It would be useful if this could be made clearer in the manuscript.

Yes, you interpreted this right. We slightly modified the wording of Sect. 2.1 to make this clearer (and, also, we think that Fig. 1 should be helpful in this regard).

GC4) For the time periods used, why only use 2019 for the last year. Would it make more sense to use e.g. 1994-2004, 2008-2012 and 2015-2019? That way you are using multiple years for a time average and investigating changes across a more distributed timeline. However, if you stick with the original time periods/years, the authors need to justify why this is the case. I'm a bit concerned of interpreting the results for just 2019 (i.e. one-year) when the other periods are at least 5-10 years in length.

We have revised the definition of the time periods to present more robust statistics of the data. We study 1994-2004 against 2011-2016 to directly compare with findings from Gaudel et al. (2020), 2017-2019 to capture changes before the COVID-19 lockdown period and 2020-2023 to study the COVID-19 and post COVID-19 lockdown period.

GC5) The authors also say that their results show a RF estimate which is 60-90% larger than e.g. Skeie et al., (2020). However, from my understanding, the Skeie et al., (2020) estimate is based

on global model simulations, while the values derived in this study are for the tropics and northern mid-latitudes. Therefore, I think this needs to be clearly stated when the comparisons are made in the manuscript.

The Reviewer is totally right and we thought this difference in our estimations and Skeie's was already implicitly clear in the manuscript. To be clearer, we added the following sentence in the paragraph discussing the comparison: "It is important to stress that the estimations for Skeie et al. (2020) are global average values, while our study is limited to the Northern Hemisphere and the tropics, which can produce biases in this inter-comparison."

GC6) When you derive the trend in RF (e.g. RHS column in Table 3), how is this done? The time period e.g. 1994-2004 covers 10-11 years, so when calculating the rate per year, how to do you determine the start and end points. For instance, when comparing 1994-2004 to 2011-2016, would that be e.g. the midpoints 1999-2013 which is 14-years? This needs to be clarified.

Yes, we take the midpoint of all time intervals. Please also note that in the revised manuscript version we now report on the $\Delta$TOC, $\Delta$LTOC, $\Delta$UTOC and RF per decade and not per year, as done in the previous version.

GC7) Discussion on Table 5 and Equations 1-3 seems a bit rushed to me (only two sentences). I believe there is an opportunity for a more detailed discussion on this. For instance, Rap et al., (2015) and Pope et al., (2024) discuss the sensitivity of the RE wrt tropospheric ozone (i.e. normalised tropospheric ozone radiative effect). How do your calculated values of RE for each time period compare with those values?

We calculated these values using the Eqs. 1-3, where RF and $\Delta$TOC are averages in the different time intervals and are made separately for the 11 regions (and then averaged to have one value only with its variability). In the revised version of the manuscript, we now mention a comparison with the results of Pope et al. (2024). As the RF sensitivity depends strongly on the latitude (see Figure 1 of Pope et al. (2024), we could not compare our results with those of Rap et al. (2015), for which the variability of the RF sensitivity with latitude is not shown.

GC8) To calculate the upper and full tropospheric column of ozone, you fix the tropopause at 11 km. The tropopause will depend heavily on latitude, so this could lead to misleading column values. I know the authors say that a chemical tropopause value of 125 ppbv of ozone is used, so stratospheric air is not influencing the column values, however, the tropopause can reach up to 15-20 km in the tropics, so you might be underestimating the tropical tropospheric column amounts and associated RF.

This is an unavoidable limitation of IAGOS observations, as they stop at about this altitude. This is mentioned in the text and in the many references about IAGOS in Sect. 2.

**Minor Comments:**

MC1) The abstract was a bit confusing. The use of 2011-2016 and 2019 becomes clearer when reading the full paper, but if only looking at the abstract, it is difficult to understand which values are associated with each later time period. Therefore, I suggest rewording this part of the abstract.

We slightly modified the wording here, so to make it clear that the starting point of tropospheric ozone trends and the associated RF is always the same (1994-2004); also, please note that now we have different periods for the ending point of the trends.

MC2) "m-2" should be superscript in many places in the abstract.
MC3) Page 2 Line 49: Pope et al., (2024) focus on satellite data between 2008 and 2017, not 2019.
Both corrected

MC4) Page 3 Lines 85-87 are unclear. Please reword.
We reworded the sentence and hope that now it is clearer

MC5) Page 5 Line 127: Instead of "aren't", I suggest "are not". This occurs a couple of times in the manuscript.
Corrected (but we did not find any more "aren't" in the text...)

MC6) Page 6 Line 135: The difference value is smaller than the uncertainty (or range). So, do you have confidence that this is an increasing UT O3 value? Secondly, what is the metric in the brackets based on? Is it an uncertainty metric or range in the data?
This is no more applicable as we changed the time intervals. In general, in parentheses is a range of the data, associated with their variability.

MC7) Page 7 Line 142. Please define VOCs in the first instance earlier in the manuscript.
Done

MC8) Figure 4: Would does Figure x refer to?
This was intended as Tab. 1; corrected

MC9) Figure 5: Suggest making the dotted lines may be dashed lines. They don't look overly clear to me. Just a suggestion though.
Figure 5 is now changed a bit and hope is clearer now.

References:

- Rap, A., Richard, N. A. D., Forster, P. M., Monks, S. A., Arnold, S. R., and Chipperfield, M. P.: Satellite constraint on the tropospheric ozone radiative effect, Geophys. Res. Lett., 42, 5074–5081, https://doi.org/10.1002/2015GL064037, 2015.
- Pope, R. J., Rap, A., Pimlott, M. A., Barret, B., Le Flochmoen, E., Kerridge, B. J., Siddans, R., Latter, B. G., Ventress, L. J., Boynard, A., Retscher, C., Feng, W., Rigby, R., Dhomse, S. S., Wespes, C., and Chipperfield, M. P.: Quantifying the tropospheric ozone radiative effect and its temporal evolution in the satellite era, Atmos. Chem. Phys., 24, 3613–3626, https://doi.org/10.5194/acp-24-3613-2024, 2024.

---

## Author Response (AR2)

Reply to the Editor and Reviewer #1 of the manuscript "Decadal tropospheric ozone radiative forcing estimations with offline radiative modelling and IAGOS aircraft observations", Sellitto et al., second review stage.

Dear Editor, dear anonymous Reviewer #1,

Many thanks for your kind words about our manuscript and our review work in the previous stage. We have implemented the second round of comments: see our point-by-point reply in the following.

Thank you very much for your commitment to the review process of our manuscript, Pasquale Sellitto on behalf of all co-authors

**Reviewer #1**

Thanks for taking carefully into account my comments and doing the additional analyses with an extended time interval (2017-2019) and the (post—COVID time interval (2000-2023). I agree with you that these analyses greatly improved the manuscript. When my comments and suggestions have not been taken into account, the authors provide meaningful arguments not to do so, so I believe the manuscript is ready for publication in ACP. I mainly have some minor or technical comments, mostly related to the (visibility of the) figures.

Thank you for your constructive criticism and these kind words.

- 1) Line 39: change to "was found to continue increasing well" Done
- 2) Line 46: change to "ground-based, ozonesonde and satellite observations" Done
- 3) Fig. 2: as already asked in my previous review report: why not extending the profiles up to 11 km (instead of 10 km)? You define the TOC and UTOC up to 11 km, so it would make sense to show the average profiles also up to 11 km!

We are very sorry, our reply to this question was somehow lost during the first review round. The reason why we did not extend the top altitude to 11 km for this figure was because the number of observations available at this altitude is not completely homogeneous across the different regions (for some regions, at this altitude we had a certain number of rejected observations due to influence of stratospheric airmasses). While this has a very minor (if any perceptible) impact on both the UT ozone trends and the associated RF estimations, we decided to stop the profile figures only at 10 km, where these small inhomogeneities are not there anymore.

4) Fig 2: I found the very hard to distinguish between the mean vertical profiles colored the grey and black on one hand, and red and dark red line on the other hand (in my printed version of the manuscript). Why not using 4 different colors here? Or increasing the contrast between grey and black and red and dark red and using dotted lines for two of them?

We changed solid grey lines to dashed grey lines (1994-2004) and we increased the contrast of the red lines (2017-2019) using a lighter shade of red. It looks like more readable now, we think.

5) Lines 145-146: "the general reduction" of UTOC in 2020-2023 seems to contrast with the

positive values between the brackets (+0.5+/-1.1 DU, +6.6+/-13.9%  $\diamond$  also add DU here). So, please specify the reference period for the reduction (I guess w.r.t. 2017-2019?) here! There was a word missing here: "...this is not the case for 2020-2023 when a general reduction of UTOC \*trends\* is found (+0.5 $\pm$ 1.1 DU, +6.6 $\pm$ 13.9%)...": we corrected the sentence and also added "DU", thanks.

6) Fig. 3: here again, if you would change the colors in Fig. 2, this would also increase the readability between the different column bars: the difference between red and dark red is very hard to distinguish!

Done, with the same colour scheme as for Fig. 2 (see comment 4 above).

7) Fig. 5: the difference between the two column bars of the different regions would be clearer (especially if they are close) if they would have an outside border line. The horizontal dotted lines that represent the uncertainty range of the solid lines are very hard to read (in my printed version). Couldn't you work e.g. with an uncertainty interval to the right of the figure (the horizontal solid lines are extended to the right and its uncertainty is marked there?).

We have added a black border line for the different column bars. We agree that the horizontal dotted lines (uncertainty interval for the solid lines) make the figure a bit busy, so we adapted a bit the suggestion and we only show them at the figure edge. Please also note that we adapted the colours of the horizontal lines to the new colours of Figs. 2-3 (lighter red).

8) Line 263: Replace with "Three time intervals" Done, thanks

---

## Author Response (AR3)

Reply to the Editor of the manuscript "Decadal tropospheric ozone radiative forcing estimations with offline radiative modelling and IAGOS aircraft observations", Sellitto et al., third review stage.

**Dear Andreas,**

I'm sorry, I did not check the length of the Abstract in the second iteration of the review of this manuscript. I have now modified it so that it has 250 words. I also added the Data Availability statement, as you suggested (thank you for the specific suggestion).

Thank you very much for your work during the review of our manuscript, Pasquale